# A Kinematic Study on the Use of Overhead Squat Exercise with Elastic Resistance on the Shoulder Kinetic Chain Approach

**DOI:** 10.3390/jfmk10010097

**Published:** 2025-03-18

**Authors:** Fagner Luiz Pacheco Salles, Augusto Gil Pascoal

**Affiliations:** 1LBMF, CIPER–Neuromechanics, Faculty of Human Kinetics, University of Lisbon, 1649-004 Lisboa, Portugal; fagnersalles@edu.ulisboa.pt; 2Alcoitão School of Health Sciences, 2649-506 Alcabideche, Portugal

**Keywords:** shoulder kinematics, trunk kinematics, rehabilitation, overhead squat, elastic resistance training

## Abstract

**Background:** The overhead squat movement involves various bodily structures, but the interaction with three-dimensional elastic resistance along the kinetic chain approach requires further understanding. **Objectives**: We aim to describe and compare scapular and trunk kinematics during an overhead squat under different external resistance conditions. **Methods**: The three-dimensional shoulder and trunk kinematics of 19 male participants were captured at 15-degree intervals, from 30 to 120 degrees, during the overhead squat movement and analyzed by phase. **Results**: Scapular posterior tilt was significantly affected by resistance during the UNLOAD phase (*p* = 0.005, η^2^ₚ = 0.26). Significant resistance-by-arm elevation interactions were found for scapular upward rotation during the LOAD phase (*p* = 0.003, η^2^ₚ = 0.19) and UNLOAD phase *(p <* 0.001, η^2^ₚ = 0.24); for scapular internal rotation during both the LOAD (*p* < 0.001, η^2^ₚ = 0.37) and UNLOAD phases (*p* = 0.006, η^2^ = 0.19); and for scapular posterior tilt during both the LOAD (*p* = 0.003, η^2^ₚ = 0.26) and the UNLOAD phases (*p* = 0.006, η^2^ₚ = 0.21). Trunk flexion/extension showed a significant effect on resistance during the LOAD phase (*p* = 0.008, η^2^ₚ = 0.24). **Conclusions**: Increasing resistance through elastic resistance significantly improves scapular kinematics via the trunk during arm elevation. This underscores the potential of the overhead squat movement as a valuable tool for assessing and treating scapular and trunk dysfunction.

## 1. Introduction

The shoulder complex enables a broad range of movements by interacting with its structures [1]. The scapulothoracic joint is central in providing stability for the humeral head [2,3,4,5] and linking the trunk to the upper limb via the kinetic chain [6]. Scapular and glenohumeral motion disruptions are commonly observed in shoulder issues and contribute to biomechanical factors in shoulder injuries [7,8].

Human movement is a complex process involving a coordinated sequence of actions across body segments. The kinetic chain (KC) describes these interconnections, with activation beginning in the lower body and progressing through the hip and trunk to coordinate upper limb movements via the scapula [6,9], which involves the sequential activation of body segments during daily activities and sports [4,5]. Movement begins in the lower body and ascends through the hip and trunk, ultimately aiding the coordination of upper limb movements via the scapula [6,10].

The squat exercise exemplifies the KC concept, with the overhead squat (OHS) being particularly challenging as it requires the scapula to remain aligned with the thoracic spine while the glenohumeral joint facilitates arm movements [11]. The KC concept has been extensively applied in shoulder rehabilitation for athletic populations [6,12]. Similarly, the OHS is frequently utilized as both an assessment and treatment tool for lower limb function. However, its effects on shoulder kinematics, as well as reference values under varying load conditions during the OHS, remain poorly understood, particularly in healthy populations.

Current shoulder rehabilitation programs incorporate a variety of therapeutic modalities [13,14], with elastic bands standing out as versatile and effective tools. Beyond strength and endurance training, elastic resistance [15,16] has been shown to enhance the range of motion [17,18], joint stability [19], and overall performance [15,16]. However, most existing approaches primarily emphasize unidirectional resistance, either in association with the KC [20,21,22,23] or independently [17,24]. Despite this, the role of elastic resistance within the KC during vertical displacement and its effects on scapular and trunk kinematics remain insufficiently understood and require further investigation. Additionally, research indicates that trunk positioning significantly influences scapular kinematics [25,26,27]. Nevertheless, the impact of incorporating elastic resistance into OHS movements on scapular and trunk kinematics remains unclear.

Thus, this study aims to describe and compare scapular and trunk kinematics during an overhead squat under different external resistance conditions. We hypothesize that (1) the OHS with external resistance will improve scapular kinematics, demonstrating better adaptation and alignment compared to the no-resistance condition, as a response to the added load; and (2) during the OHS, the trunk exhibits adaptive responses to maintain stability and effectively support scapular motion during external rotation compared to the no-resistance condition. This study aimed to clarify how a specific exercise, combined with a kinetic chain resistance condition, influences scapular kinematics and its potential benefits in treating shoulder issues, particularly those related to scapular dysfunction.

## 2. Materials and Methods

### 2.1. Participants

This study involved nineteen healthy young adults, aged between 20 and 23 years [age = mean (SD) 21.9 (3.7) years; height = 175.7 (6.2) cm; weight = 80 (13.9) kg; and Body Mass Index (BMI) = 25.2 (4.2) kg/m^2^]. Participants were recruited through convenience sampling. Inclusion criteria required participants to be free from shoulder or neck issues for the past 6 months and to have no history of shoulder fractures or surgeries. Participants underwent the lunge [28,29] and step-down tests [30] to ensure no lower limb kinetic chain alterations. Participants were then informed of the details of study participation and provided written consent approved by the University’s Institutional Review Board (CEIFMH n°: 45/2021; Approval Date: 3 November 2021), and all study procedures were conducted according to the Declaration of Helsinki and Human Subjects Research Guidelines.

#### Sample Size Estimation

An “a priori” sample size calculation was conducted using G*Power (version 3.1.9.2), drawing on insights from the studies conducted by Hotta et al. [31] and Bench et al. [32] studies, which investigated scapular motor control exercises and elastic resistance. For the sample size calculation in this study, we employed an effect size of 0.175, a significance level of 0.05, a power (1 − β error probability) of 0.80, a nonsphericity correction ε of 1.0 (ANOVA–repeated-measures test), and a partial _Ƞ_^2^ of 0.03. The calculation revealed that a total sample size of at least 18 participants was required.

### 2.2. Instrumentation and Data Collection

A 6-degree-of-freedom electromagnetic tracking device (“FASTRACK” by Polhemus, Colchester, VT, USA), with the MotionMonitor™ software (Motion Monitor V9, Innovative Sports Training, Chicago, IL, USA) was used to record the three-dimensional (3D) position and orientation of the thorax and shoulder tracked at a sampling rate of 30 Hz. The system has a reported root mean square (RMS) accuracy of 0.3–0.8 mm for position and 0.15° for orientation when operating within a 76 cm distance between the source and the sensor (SPACE FASTRAK User’s Manual, Revision F. Colchester, VT; Polhemus Inc.; 1993) [33]. This system included a transmitter and four sensors. The sensors were attached to the following bony landmarks of the participants using adhesive tape: the thorax and right acromion. A customized cuff secured the right humeral sensor (Figure 1).

For the segment orientation description, local coordinate systems (LCSs) were defined by aligning each sensor with specific bony landmarks, as recommended by the International Society of Biomechanics [34]. A fourth sensor, mounted on a pointer, was used to manually digitize anatomical landmarks on the thorax, humerus, and scapula. These landmarks were palpated and digitized to transform the sensor data from the global coordinate system (the transmitter) to an anatomically based LCS. Measurements were taken with participants sitting (90° flexion at the hips and knees) with their arms along the thorax, elbows flexed, and hands resting on their knees. The glenohumeral rotation center was estimated with a least squares algorithm, defined as the point that moved the least during several passive arm circumduction movements [35]. By combining the LCSs constructed from these anatomical landmarks and the sensor motions, both segment and joint rotations could be calculated [34].

In this study, the orientations of the scapula and humerus relative to the thorax were analyzed using three-dimensional (3D) bone kinematics described by Euler angles. Scapular rotations were represented using the Euler angle sequence (Y-X′-Z″) as scapular internal (positive) and external rotation, upward (positive) and downward rotation, and anterior (negative) and posterior scapular tilts. The humerus rotations were defined as thoracohumeral angles using a Euler angle sequence (Y-X′-Y″), resulting in the plane of elevation as anterior (positive) and posterior, elevation (negative) and depression, and axial rotation as external rotation (negative) and internal rotation. To provide a more intuitive description of scapular and humeral motion around the X′-axis, the rotations were multiplied by −1. Thus, positive values now represent the upward rotation of the scapula and arm elevation. Thorax rotations were represented using the Euler angle sequence (Y-X′-Z) relative to the global reference coordinate system, which describes the left (positive) and right rotation, the left (negative) and right flexion, and flexion (positive) and extension.

### 2.3. Task Procedures

Following the digitization process, kinematic data were collected during five trials of a bilateral upper limb elevation–depression action, performed simultaneously with a squat–return motion. In these trials, participants stood upright with their arms hanging and relaxed at their sides. They then began the task by raising their arms (flexion) synchronously with the squat motion, which involves flexing their thighs and knees. After reaching the maximum squat depth, participants returned to the initial position by extending their arms, thighs, and knees (Figure 2). The task was performed under three conditions: no elastic resistance (R00); elastic resistance applied from the feet to the upper limbs (R01); and elastic resistance applied between the upper and lower limbs (R02).

The exercise was divided into two phases: the LOAD phase and the UNLOAD phase. During the LOAD phase, arm movement occurred against external resistance. In the UNLOAD phase, the arms returned to the starting position with the assistance of gravity and/or the elastic band. In the R00 condition, gravity was the only external force, while in the R01 and R02 conditions, resistance was the combined effect of gravity and the tension of the elastic band (TheraBand^®^, Performance Health, Akron, OH, USA).

Participants followed standardized verbal and visual instructions to ensure consistency throughout the exercise [36]. They were instructed to stand with their feet aligned with the anterior superior iliac spine and maintain a straight back throughout the movement; the squat should be performed consistently during both the downward and upward phases. Participants were asked to descend as low as possible while squatting and to raise their arms during the descent, keeping them aligned with the trunk at the end of the movement. Upon ascent, they were instructed to return their arms to the starting position.

### 2.4. Statistical Analysis

#### 2.4.1. Variables

Data were analyzed using IBM SPSS software, version 28 (IBM Corp., Armonk, NY, USA). The independent variables included TheraBand^®^ type (LT-B vs. HT-B), the resistance conditions (R00 vs. R01 vs. R02), and arm elevation angles.

To examine the effect of TheraBand type, the sample was randomly divided into two groups based on the TheraBand^®^ used: the LT-Blue group and the HT-Black group. The randomization process was conducted digitally using a computer-generated algorithm to ensure unbiased group assignment. Each participant was assigned to one of the TheraBand groups to control potential variability in resistance levels and to enable a comparative analysis of kinematic effects between the Low-Tension and High-Tension bands.

Dependent variables were scapular and trunk 3D positions recorded over five repetitions for each condition. Mean repetition values were used for analysis. Scapular and trunk kinematics were evaluated across seven arm elevation positions (30° to 120° in 15° increments). Separate analyses were conducted for each scapular rotation and trunk flexion/extension by phase (LOAD and UNLOAD).

#### 2.4.2. Statistical Test

The normality of the dependent variables was assessed using the Shapiro–Wilk test and Z-scores, which were calculated by dividing the skewness and kurtosis values by their standard errors, as described by Mishra, et al. [37]. A Z-score greater than 1.96 indicated that the data approximated a normal distribution.

A repeated-measures analysis of variance (ANOVA) was performed to evaluate the effects of resistance on 3D kinematics of the shoulder and trunk. This analysis included two within-subject factors: resistance conditions (R00 vs. R01 vs. R02) and arm elevation angles (seven positions ranging from 30° to 120° in 15° increments). Additionally, a between-subject factor was incorporated: TheraBand type (Low Tension, TheraBand Blue vs. High Tension, TheraBand Black). The within-subject factors allowed for the examination of how different resistance conditions and arm positions influenced kinematic outcomes across the same individuals, while the between-subject factor enabled the comparison [38] of kinematic data between groups using different TheraBand types.

Statistical significance was set at *p* < 0.05. Mauchly’s test was used to assess the sphericity of the data. In case of violations, the Greenhouse–Geisser correction was applied. Post hoc comparisons using Bonferroni-adjusted *t*-tests were conducted to identify significant differences when significant ANOVA effects were observed. Partial eta squared (η^2^ₚ) was used as the effect size measure, with thresholds defined as follows: 0.01 for a small effect, 0.06 for a medium effect, and 0.14 for a large effect [39].

## 3. Results

### 3.1. Effect of Types of Elastic Bands (TheraBand^®^)

The results showed no significant differences between LT-B and HT-B (*p* > 0.05) for scapular and trunk kinematics.

### 3.2. Effect of Elastic Resistance

#### 3.2.1. Scapular Internal/External Rotation

There was no statistically significant main effect of resistance on scapular internal/external rotation during the LOAD phase (*p* = 0.088, η^2^_p_ = 0.13) or UNLOAD phase (*p* = 0.131, η^2^_p_ = 0.11). However, a significant interaction between resistance and arm elevation angle was observed during both phases (LOAD phase: *p* < 0.001, η^2^_p_ = 0.37; UNLOAD phase: *p* = 0.006, η^2^_p_ = 0.19).

During the LOAD phase, the scapula was positioned in more internal rotation in the R01 and R02 conditions compared to the R00 condition at 30° and the R01 condition compared to the R00 at 45° arm elevation angles. Similarly, during the UNLOAD phase, the scapula was in more internal rotation in the R02 condition compared to the R01 condition range of 30–60° arm elevation angles (Figure 3; Table 1). These findings suggest that external resistance induces specific changes in scapular internal rotation, with variations depending on the arm elevation angle.

#### 3.2.2. Scapular Upward/Downward Rotation

For scapular upward/downward rotation, no statistically significant main effect of resistance was found during either the LOAD phase (*p* = 0.295, η^2^_p_ = 0.06) or the UNLOAD phase (*p* = 0.118, η^2^_p_ = 0.11). However, a significant interaction was found between resistance and arm elevation during both phases (LOAD phase: *p* = 0.003, η^2^_p_ = 0.19; UNLOAD phase: *p* < 0.001, η^2^_p_ = 0.24). This indicates that the effect of elastic resistance on scapular upward rotation depends on the arm elevation angle.

During the LOAD phase, the scapula assumed a more upward position in the R00 condition compared to the R01, at both 45° and 60° arm elevation angles. Conversely, during the UNLOAD phase, the scapula assumed a more upward position in the R01 condition compared to the R00 condition at 30° and 45° arm elevation angles. Additionally, a more scapular upward position was also observed in the R01 condition compared to the R02 condition during the UNLOAD phase at 45° arm elevation angles (Figure 3; Table 2).

#### 3.2.3. Scapular Anterior/Posterior Tilt Rotation

For scapular anterior/posterior tilt, a statistically significant main effect of resistance was observed during the UNLOAD phase (*p* = 0.005, η^2^_p_ = 0.26). Pairwise comparisons revealed a higher posterior tilt in the R02 condition compared to the R01 condition (*p* = 0.002). Additionally, a significant interaction between resistance and arm elevation was found in both phases (LOAD phase: *p* = 0.003, η^2^_p_ = 0.26; UNLOAD phase: *p* = 0.006, η^2^_p_ = 0.21).

At the minimum arm elevation angle (30°), the scapula exhibited greater posterior tilting across all resistance conditions, with the effect being more pronounced in the R00 and R01 conditions. At 60° arm elevation, the scapula shifted to a more anterior tilt. By the maximum arm elevation angle (120°), the scapula returned to a position like the 30° elevation, showing a more posterior tilt. This pattern of scapular tilt was consistent across all resistance conditions, highlighting its prominence, particularly in the R00 and R01 conditions (Figure 3; Table 3).

#### 3.2.4. Trunk Flexion/Extension

There was a statistically significant main effect of resistance observed during the LOAD phase (*p* = 0.008, η^2^_p_ = 0.24). Pairwise comparisons revealed a higher flexion in the R01 condition than in the R00 condition (*p* = 0.024). Additionally, no significant interaction between resistance and arm elevation was found in both phases (LOAD phase: *p* = 0.178, η^2^_p_ = 0.09; UNLOAD phase: *p* = 0.496, η^2^_p_ = 0.04) (Figure 4).

## 4. Discussion

Our results indicated that resistance had varying effects on scapular kinematics depending on the phase of movement and the angle of arm elevation. However, no differences were found for the trunk kinematics. Additionally, the resistance group demonstrated that the band’s intensity did not influence shoulder and trunk kinematics. The slight difference in tension between the bands likely accounts for this outcome [24,40,41]. Thus, the resistance within the kinetic chain drives scapular adaptations, regardless of the specific tension band employed, emphasizing the band’s overall resistance impact.

This study found that adding external resistance significantly impacted scapular kinematics, particularly from 30° to 60° of arm elevation during both the LOAD and UNLOAD phases. These interactions indicate that resistance’s effect on scapular kinematics depends on the arm’s elevation angle, highlighting that the scapula’s response to added load is not uniform across different phases of the movement. The observed changes include variations in scapular internal rotation, upward rotation, and posterior tilt, supporting the notion that the scapula alters its kinematics in response to external resistance. These findings for scapular rotation align with previous research involving elastic resistance in the sagittal plane [1,42]. Above 60° of arm elevation, the scapular movement at the scapulothoracic joint must be coordinated with the humerus movement at the glenohumeral joint (scapulohumeral rhythm) to provide a stable base for a better centralization of the humeral head, contributing to and facilitating overhead movements [2,5]. Individuals with shoulder pain exhibit different scapular rotation patterns compared to both healthy individuals and those in the present study [43]. Additionally, scapular kinematics beyond 100° of arm elevation [44] play a critical role in maintaining adequate subacromial distances, thereby preventing the narrowing of these distances [45]. Thus, except for the LOAD phase during upward rotation, the scapular kinematics changes with resistance between 30° and 60° of arm elevation reinforce the concept that the scapula is influenced by the kinetic chain, adjusting its position to ensure the continuity and effectiveness of arm movements.

Additionally, the results showed a higher posterior tilt of the scapula under R02 resistance conditions during the UNLOAD phase than R00 or R01. This finding reflects how the scapula adjusts its tilt to manage the load effectively and maintain stability with the thoracic spine during movement [6,11]. Therefore, the findings of this study suggest that the OHS is a valuable tool for assessing scapular behavior within the kinetic chain. Thus, the application of external resistance alters scapular kinematics compared to the no-resistance condition, providing support for Hypothesis 1.

The study supports the idea that the scapula uses a dynamic neuromuscular strategy to optimize its position during the OHS. The observed changes in scapular rotation and tilt, particularly between 30° and 45° of arm elevation, demonstrate how the scapula adjusts its kinematics to adapt to external resistance. This adaptive neuromuscular response is a key aspect of how the body deals with added load, further validating the hypothesis. The interaction between the gluteus maximus and the contralateral latissimus dorsi [46,47] (Carvalhais et al., 2013; Mohamed et al., 2022) contributes to the concept of the kinetic chain and reinforces the importance of considering the entire kinetic chain’s functional integrity when assessing and treating musculoskeletal conditions, including shoulder issues. Furthermore, changes in the behavior of the scapula and humerus at the end of the LOAD phase and the beginning of the UNLOAD phase suggest that resistance-induced changes enhance proprioceptive feedback, influencing scapular movement relative to the humerus [48].

The results indicated that the trunk plays a crucial role as a stable base for upper quadrant movements, ensuring the proper alignment of the scapula with the thoracic spine during arm elevation. Throughout the OHS exercise, trunk-maintained stability is essential for effective force production, transfer, and control to the distal segments [6]. Although the trunk had a main effect on resistance during the LOAD phase, no interactions were observed with different resistance conditions by phases. This consistency underscores the trunk’s role in providing a stable foundation for scapular motion through core stability [6] regardless of the resistance applied. Research has demonstrated that trunk positioning significantly influences the kinematics of both the scapula and the humerus [25,27]. However, these studies did not use any resistance, with the kinematic movement being assessed with the upper limb free of load. Thus, trunk stability during the movement, irrespective of the resistance condition, played a crucial role in maintaining proper scapular alignment, thereby supporting Hypothesis 2.

The findings showed that squat movements, including the OHS, are valuable for assessing and ensuring lower limb stability [49,50,51,52,53]. This integration of lower limb stability with trunk stability reinforces the concept that the entire kinetic chain must function cohesively to support scapular motion during exercises like the OHS. Additionally, trunk instability can lead to increased tension in the latissimus dorsi, affecting scapular rotations—particularly upward rotation and posterior tilt—during arm elevation, especially in individuals with chronic low back pain [47,54]. The kinetic chain’s influence is particularly evident in the R02 condition, where an additional stimulus for hip abduction movement during both phases is notably prominent in the internal rotation and posterior tilt during the UNLOAD phase. The gluteus maximus demonstrates significant activation during the squat [55], which is further enhanced by hip abduction. The energy transfer along the kinetic chain occurs in an ascending direction, passing through the hip and trunk, leading to coordinated movements in the upper limb [6,10]. During the OHS return phase, the body counteracts gravity, aiding the upper limb in supporting the arm with resistance during the UNLOAD phase. This pattern is especially observed within the posterior tilt range of 45–105°, suggesting that resistance promotes a coordinated motor response during the eccentric phase to maintain an optimal instantaneous center of rotation for the humeral head.

In clinical practice, the use of TheraBand^®^ with different tension levels (Low and High) shows no significant differences in scapular and trunk kinematics during OHS. However, resistance exercises can influence scapular internal rotation, upward rotation, and anterior/posterior tilt, especially at varying arm elevation angles. This highlights the importance of customizing resistance levels based on individual patient needs to optimize shoulder rehabilitation outcomes.

This study had some limitations. First, the sample size was relatively small, which may limit the generalizability of the findings. Second, the study design did not account for individual variations in shoulder anatomy and biomechanics, which could influence the results. Third, this study focused solely on healthy men, so the findings may not directly apply to clinical populations with shoulder pathologies or to differences between gender. Additionally, the resistance levels used in this study were limited to three specific conditions, which may not represent the full range of therapeutic options available.

Future research directions: Future studies should aim to include larger and more diverse samples to enhance the generalizability of the results. Research should also explore the effects of elastic resistance on individuals with different shoulder pathologies to understand its therapeutic potential in clinical settings better. Additionally, examining a wider range of resistance levels and incorporating longitudinal designs to assess long-term outcomes would provide more comprehensive insights. Investigating the integration of elastic resistance exercises with other rehabilitation modalities, such as manual therapy or neuromuscular training, could further optimize shoulder rehabilitation protocols.

## 5. Conclusions

The results of this study indicate that the elastic resistance, combined with specific arm elevation angles, significantly influences scapular kinematics across different phases. These findings emphasize the importance of carefully selecting resistance levels in therapeutic exercises to optimize scapular positioning and function, thus ensuring comprehensive kinetic chain integrity. However, many factors related to the kinetic chain and elastic resistance remain underexplored, and more detailed studies are required in the future.

## Figures and Tables

**Figure 1 jfmk-10-00097-f001:**
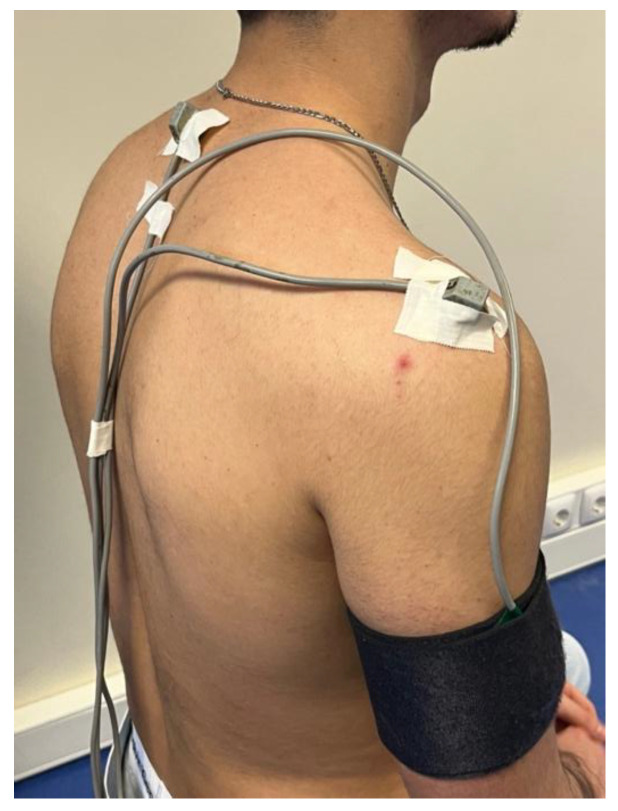
Kinematics set-up.

**Figure 2 jfmk-10-00097-f002:**
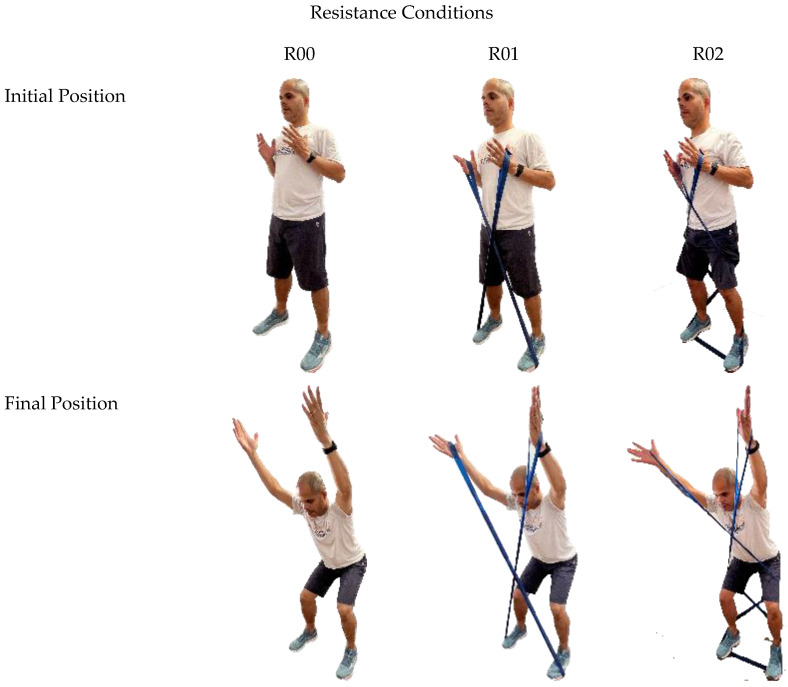
Initial and final positions of the overhead squat exercise under three resistance conditions: no elastic resistance (R00); elastic resistance from feet to upper limbs (R01); and elastic resistance between upper and lower limbs (R02).

**Figure 3 jfmk-10-00097-f003:**
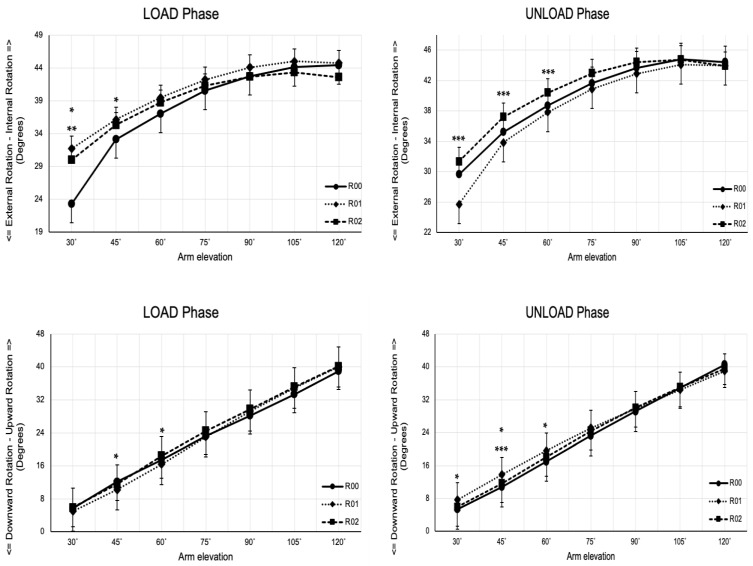
Scapulothoracic rotation across selected angles of arm position (angles) during the LOAD and UNLOAD phases of continuous arm elevation exercise. Data are mean ± Standard Error of the Mean. Significant resistance-by-arm elevation interactions are denoted as * R00 vs. R01, ** R00 vs. R02, and *** R01 vs. R02.

**Figure 4 jfmk-10-00097-f004:**
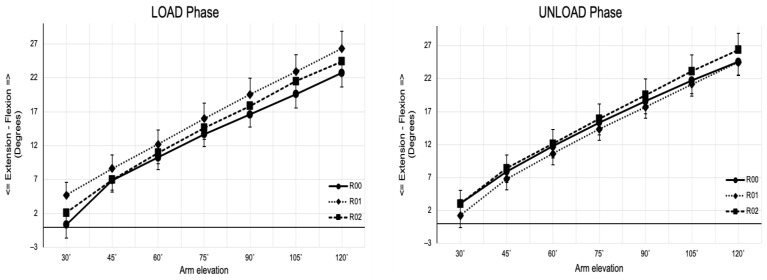
Trunk rotation across selected angles of arm position (angles) during the LOAD and UNLOAD phases of continuous arm elevation exercise. Data are mean ± Standard Error of the Mean.

**Table 1 jfmk-10-00097-t001:** Interaction effects of elastic resistance, arm elevation angle, and phase on scapular internal/external rotation.

Arm Elevation	Phase	Resistance	Comparison	Difference (SD)	95% CI	*p*-Value
30°	LOAD	R01 vs. R00	Internal position in R01	8.41 (8.19)	15.06 to 58.27	0.001
30°	LOAD	R02 vs. R00	Internal position in R02	6.72 (10.15)	2.51 to 56.09	0.029
45°	LOAD	R01 vs. R00	Internal position in R01	3.03 (4.37)	1.67 to 24.72	0.022
30°	UNLOAD	R02 vs. R01	Internal position in R02	5.65 (6.13)	8.45 to 40.83	0.002
45°	UNLOAD	R02 vs. R01	Internal position in R02	3.38 (4.25)	3.52 to 25.94	0.008
60°	UNLOAD	R02 vs. R01	Internal position in R02	2.56 (3.96)	0.70 to 21.61	0.034

Resistance condition: R00 = no resistance, R01 = one resistance and R02 = two resistance.

**Table 2 jfmk-10-00097-t002:** Interaction effects of elastic resistance, arm elevation angle, and phase on Scapular upward/downward rotation.

Arm Elevation	Phase	Resistance	Comparison	Difference (SD)	95% CI	*p*-Value
45°	LOAD	R00 vs. R01	Upward position in R00	1.91 (1.56)	4.20 to 12.42	0.000
60°	LOAD	R00 vs. R01	Upward position in R00	1.07 (1.68)	0.24 to 9.11	0.037
30°	UNLOAD	R01 vs. R00	Upward position in R01	2.28 (3.16)	1.58 to 18.29	0.017
45°	UNLOAD	R01 vs. R00	Upward position in R01	3.01 (3.40)	4.15 to 22.11	0.003
45°	UNLOAD	R01 vs. R02	Upward position in R01	2.15 (2.83)	1.90 to 16.83	0.012
60°	UNLOAD	R01 vs. R00	Upward position in R01	2.67 (3.23)	3.13 to 20.17	0.006

Resistance condition: R00 = no resistance, R01 = one resistance and R02 = two resistance.

**Table 3 jfmk-10-00097-t003:** Interaction effects of elastic resistance, arm elevation angle, and phase on scapular anterior/posterior tilt rotation.

Arm Elevation	Phase	Resistance	Comparison	Difference (SD)	95% CI	*p*-Value
30°	LOAD	R01 vs. R00	Posterior Tilt in R01	1.84 (2.81)	0.61 to 15.42	0.031
30°	LOAD	R01 vs. R02	Posterior Tilt in R01	2.45 (2.92)	2.94 to 18.38	0.006
45°	LOAD	R00 vs. R02	Posterior Tilt in R00	1.88 (2.99)	0.29 to 16.07	0.041
45°	LOAD	R01 vs. R02	Posterior Tilt in R01	2.67 (3.41)	2.64 to 20.66	0.009
45°	UNLOAD	R00 vs. R01	Posterior Tilt in R00	2.81 (2.84)	4.73 to 19.73	0.001
45°	UNLOAD	R02 vs. R01	Posterior Tilt in R02	2.17 (2.77)	2.15 to 16.77	0.009
60°	UNLOAD	R00 vs. R01	Posterior Tilt in R00	2.98 (3.23)	4.46 to 21.48	0.002
60°	UNLOAD	R02 vs. R01	Posterior Tilt in R02	2.80 (2.85)	4.66 to 19.72	0.001
75°	UNLOAD	R00 vs. R01	Posterior Tilt in R00	2.30 (3.67)	0.33 to 19.69	0.041
75°	UNLOAD	R02 vs. R01	Posterior Tilt in R02	2.83 (2.81)	4.93 to 19.78	0.001
90°	UNLOAD	R02 vs. R01	Posterior Tilt in R02	2.81 (2.73)	5.03 to 19.43	0.001
105°	UNLOAD	R02 vs. R01	Posterior Tilt in R02	2.42 (2.77)	3.23 to 17.85	0.004

Resistance condition: R00 = no resistance, R01 = one resistance and R02 = two resistance.

## Data Availability

Data are available upon request.

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
