# Peer review of "A Kinematic Study on the Use of Overhead Squat Exercise with Elastic Resistance on the Shoulder Kinetic Chain Approach"

_jfmk, 2025, doi:10.3390/jfmk10010097_

Round 1

Reviewer 1 Report

Comments and Suggestions for Authors

Thank you for the opportunity to review your manuscript, “A Kinematic Study on the Use of Overhead Squats Exercise 2 with Elastic Resistance for Shoulder Rehabilitation.”

It is an interesting work. However, there are aspects that I believe should be improved:

  1. The main limitation of the study is that the subjects were very young and did not have any shoulder pathology. The authors acknowledge that the results cannot be extrapolated directly to populations with shoulder pathology. In my opinion, the authors should modify the title of the paper. It should not include the term "for shoulder rehabilitation."
  2. As for the study participants, the authors included 19 healthy young subjects. They make no reference to sex. Differences between men and women in terms of elasticity and even strength are well known. I think the authors should clarify and include this aspect among the limitations of the study.
  3. The results mention changes in scapular and trunk kinematics; however, these are not compared with clinical reference values or pathological populations. This aspect should be more clearly addressed in the study's limitations.
  4. Regarding the interpretation of the data:
    1. It is concluded that "increasing resistance with elastic bands improves scapular kinematics," but this claim is not fully supported in the discussion. Additional references are needed to explain how these changes impact shoulder injury rehabilitation.
    2. The relationship with the kinetic chain is mentioned; however, without a detailed muscular assessment (such as electromyography), it is difficult to substantiate certain statements regarding stability and neuromuscular activation.

Author Response

Comments 1: [It is an interesting work. However, there are aspects that I believe should be improved:

  1. The main limitation of the study is that the subjects were very young and did not have any shoulder pathology. The authors acknowledge that the results cannot be extrapolated directly to populations with shoulder pathology. In my opinion, the authors should modify the title of the paper. It should not include the term "for shoulder rehabilitation."
  2. As for the study participants, the authors included 19 healthy young subjects. They make no reference to sex. Differences between men and women in terms of elasticity and even strength are well known. I think the authors should clarify and include this aspect among the limitations of the study.
  3. The results mention changes in scapular and trunk kinematics; however, these are not compared with clinical reference values or pathological populations. This aspect should be more clearly addressed in the study's limitations.
  4. Regarding the interpretation of the data:
    1. It is concluded that "increasing resistance with elastic bands improves scapular kinematics," but this claim is not fully supported in the discussion. Additional references are needed to explain how these changes impact shoulder injury rehabilitation.
    2. The relationship with the kinetic chain is mentioned; however, without a detailed muscular assessment (such as electromyography), it is difficult to substantiate certain statements regarding stability and neuromuscular activation.]

Response 1: Thank you for pointing this out. We agree with this comment. Therefore, we have.[ made suggestion on title, this change can be found – page number 1]

“[A Kinematic Study on the Use of Overhead Squats Exercise with Elastic Resistance on Shoulder Kinetic Chain Approach]”

Response 2: Thank you for pointing this out. We agree with this comment. Therefore, we have.[made a change in study limitation. Mention exactly where in the revised manuscript this change can be found – page number 13, paragraph 1, and line 347.]

“[Third, the study focused solely on healthy men, so the findings may not directly apply to clinical populations with shoulder pathologies or to differences between gender.]”

Response 3: Thank you for pointing this out. We agree with this comment. Therefore, we have.[ The aim of this study was more focused on describing and comparing OHS with resistance arrangements. But we added an excerpt to show the difference between our result and another study. Mention exactly where in the revised manuscript this change can be found – page number 11, paragraph 2nd, and line 281.]

“[Individuals with shoulder pain exhibit different scapular rotation patterns compared to both healthy individuals and those in the present study.]”

Response 4: Thank you for pointing this out. We disagree with this comment. [R1. The articles referenced in the introduction and discussion already provide a solid foundation for explaining our results. Additionally, the use of elastic resistance integrated into the kinetic chain remains an underexplored topic, and our study contributes to expanding this body of knowledge.

R2. Regarding the kinetic chain, the studies we cited helped us explain the observed behavior. We acknowledge that EMG analysis would provide further confirmation. The EMG data related to this study are currently being processed and are expected to be published soon.]

Reviewer 2 Report

Comments and Suggestions for Authors

This study aims to assess the kinematics of overhead squats with elastic resistance for shoulder rehabilitation. It is intriguing, relevant, and methodologically well-executed. I don’t usually have the opportunity to read and review a decently executed and nicely written paper such as this one. Some minor improvements might further improve this already good manuscript.

Keywords:

  1. Please avoid repeating words from the title page in keywords.

Introduction:

  1. Overall, the introduction is well-structured and written. It is relevant and thorough, with adequate rationale. Some minor modifications should be applied:
    1. Please consider adding the significance of the study after the hypotheses.

Methods:

  1. The method section is also well-presented and detailed enough to secure replication. Some minor comments should be considered:
    1. I would suggest splitting the statistical analysis section into two sections for better understanding. In particular, paragraphs 2 and 3 could be in the section on Variables, while the rest can remain in the Statistical analysis chapter.
    2. Please add the interpretation of the effect size (I see it is calculated and shown in the results) in the Statistical analysis chapter, as well as the adequate reference. Also, be aware that many authors mistakenly interpret partial eta squared as eta squared.

Results:

  1. Results are presented clearly and in a scientific manner. Tables and graphs are visually well-presented and clearly described.

Discussion:

  1. The results were thoroughly elaborated in the discussion and adequately referenced, with acknowledged limitations. Some minor modifications should be applied:
    1. Overall, the discussion could be shortened to be more concise and easier to read and follow.
    2. Future directions might be more suitable for the conclusion chapter.

Author Response

Comments 1: [Keywords:

  1. Please avoid repeating words from the title page in keywords.

Introduction:

  1. Overall, the introduction is well-structured and written. It is relevant and thorough, with adequate rationale. Some minor modifications should be applied:
    1. Please consider adding the significance of the study after the hypotheses.

Methods:

  1. The method section is also well-presented and detailed enough to secure replication. Some minor comments should be considered:
    1. I would suggest splitting the statistical analysis section into two sections for better understanding. In particular, paragraphs 2 and 3 could be in the section on Variables, while the rest can remain in the Statistical analysis chapter.
    2. Please add the interpretation of the effect size (I see it is calculated and shown in the results) in the Statistical analysis chapter, as well as the adequate reference. Also, be aware that many authors mistakenly interpret partial eta squared as eta squared.

Results:

  1. Results are presented clearly and in a scientific manner. Tables and graphs are visually well-presented and clearly described.

Discussion:

  1. The results were thoroughly elaborated in the discussion and adequately referenced, with acknowledged limitations. Some minor modifications should be applied:
    1. Overall, the discussion could be shortened to be more concise and easier to read and follow.
    2. Future directions might be more suitable for the conclusion chapter.]

Response 1: Thank you for pointing this out. We agree with this comment. Therefore, [we understand the importance of avoiding repetition of title words in the keywords, we believe including 'overhead squat' is essential for optimizing article visibility in search engines.]

Response 2: Thank you for pointing this out. We agree with this comment. Therefore, we have.[made a change in study limitation. Mention exactly where in the revised manuscript this change can be found – page number 2, paragraph 3, and line 68.]

“[While we understand the importance of avoiding repetition of title words in the keywords, we believe including 'overhead squat' is essential for optimizing article visibility in search engines.]”

Response 3: Thank you for pointing this out. We agree with this comment. Therefore, we have.[reorganized the statistical analysis as you suggested. Mention exactly where in the revised manuscript this change can be found – page number 6, paragraph 2nd, and line 161.]

“[2.4. Statistical Analysis

2.4.1. Variables

Data were analyzed using IBM SPSS software, version 28 (IBM Corp., Armonk, NY, USA). The independent variables included TheraBand® type (LT-B vs. HT-B), the resistance conditions (R00 vs. R01 vs. R02), and arm elevation angles.

To examine the effect of TheraBand type, the sample was randomly divided into two groups based on the TheraBand® used: the LT-Blue group and the HT-Black group. The randomization process was conducted digitally using a computer-generated algorithm to ensure unbiased group assignment. Each participant was assigned to one of the TheraBand groups to control potential variability in resistance levels and to enable a comparative analysis of kinematic effects between the Low-Tension and High-Tension bands.

Dependent variables were scapular and trunk 3D positions recorded over five repetitions for each condition. Mean repetition values were used for analysis. Scapular and trunk kinematics were evaluated across seven arm elevation positions (30° to 120°, in 15° increments). Separate analyses were conducted for each scapular rotation and trunk flexion/extension by phase (LOAD and UNLOAD).

2.4.2. Statistical Analysis

The normality of the dependent variables was assessed using the Shapiro-Wilk test and Z-scores, which were calculated by dividing the skewness and kurtosis values by their standard errors, as described by Mishra, Pandey, Singh, Gupta, Sahu and Keshri 36. A Z-score greater than 1.96 indicated that the data approximated a normal distribution.

A repeated measures analysis of variance (ANOVA) was performed to evaluate the effects of resistance on 3D kinematics of the shoulder and trunk. This analysis included two within-subject factors: resistance conditions (R00 vs. R01 vs. R02) and arm elevation angles (seven positions ranging from 30° to 120° in 15° increments). Additionally, a between-subject factor was incorporated: TheraBand type (Low Tension, TheraBand Blue vs. High Tension, TheraBand Black). The within-subject factors allowed for the examination of how different resistance conditions and arm positions influenced kinematic outcomes across the same individuals, while the between-subject factor enabled the comparison of kinematic data between groups using different TheraBand types.

Statistical significance was set at P < 0.05. Mauchly’s test was used to assess the sphericity of the data. In case of violation, the Greenhouse-Geisser correction was applied. Post hoc comparisons using Bonferroni-adjusted t-tests were conducted to identify significant differences when significant ANOVA effects were observed. Partial eta squared (η²â‚š) was used as the effect size measure, with thresholds defined as follows: 0.01 for a small effect, 0.06 for a medium effect, and 0.14 for a large effect 37.]”

Response 4: Thank you for pointing this out. We disagree with this comment. Therefore, [we understand that the discussion has become somewhat lengthy; however, it effectively explains this new approach to using elastic resistance. Additionally, keeping the questions for future studies at the end of the discussion maintains the scientific rigor.]